# Behavioural and socio-ecological factors that influence access and utilisation of health services by young people living in rural KwaZulu-Natal, South Africa: Implications for intervention

**Nothando Ngwenya**[1]*, **Busisiwe Nkosi**[1], **Lerato S. Mchunu**[1], **Jane Ferguson**[1,2], **Janet Seeley**[1,2], **Aoife M. Doyle**[2]

**1** Africa Health Research Institute, KwaZulu-Natal, South Africa, **2** London School of Hygiene and Tropical Medicine, London, United Kingdom

* Nothando.ngwenya@ahri.org

**Data Availability Statement:** Data cannot be shared publicly because of confidentiality and

## Abstract

### Background

Young people's health service utilisation (the number accessing a facility) has been the focus of guidelines and health systems strengthening policies. This is due to young people being at an increased health risk because of inequitable access and utilisation of health services, which is more pronounced in rural settings with limited service availability. This is a major concern as globally, youth constitute a considerable and increasing part of the population in Sub-Saharan Africa.

### Objective

The objective of this paper is to present a comprehensive approach for the exploration of health service utilisation by young people in rural KwaZulu-Natal, South Africa. We examined barriers and facilitators conceptualised by the constructs of the Theory of Planned Behaviour, framed within a socio-ecological model.

### Methods

Data were collected in January to June 2017 from two sites using in-depth interviews, spiral transect walks and community mapping with young people (aged 10 to 24 years), primary care health providers, school health professionals, community stakeholders and young people's parents.

### Results

Socio-ecological and behavioural factors influenced young people's intention to use services. Barriers included perceived negative attitudes of health providers and perceived poor staff competencies. Facilitators included an appreciation of receiving health education and assumed improved health. At social and community levels, normative beliefs hindered

potential breach. Data are available from the AHRI Research Data Management committee (contact via RDMServiceDesk@ahri.org) for researchers who meet the criteria for access to confidential data.

**Funding:** Financial support for this research was provided by ViiV Healthcare's Positive Action for Adolescents Programme Grant number ITCRZF51; the Wellcome Trust with core funding for AHRI (082384/Z/07/Z) and joint funding under the UK Medical Research Council (MRC)/ UK Department for International Development (DFID) Concordat agreement which is supported by the European Union under the EDCTP2 programme (reference MR/K012126/1) with AD receiving support (G0700837). The funders had no role in study design, data collection and analysis, decision to publish, or preparation of the manuscript.

**Competing interests:** The authors have declared that no competing interests exist.

young people from utilising services as they feared stigmatisation and gossip. At a public policy level, structural elements had a disempowering effect as the physical layout of the clinics hindered utilisation, limited resources influenced staffing, and facility opening times were not convenient for school goers.

## Conclusion

We suggest that to fully appreciate the complexity of health service utilisation, it is necessary to not only consider factors and processes relevant to the individual, but also acknowledge and act upon, the disjuncture between community level cultural values, norms and national policies.

## Introduction

The past three decades have seen increased public health interest in promoting health service utilisation for young people (YP), a population who often have relatively low uptake of services. Health service utilisation by young people (YP) aged 10 to 24 is especially pertinent in sub-Saharan Africa (SSA). Statistics indicate that the number of young people in this age group is rapidly increasing, with a high burden of disease accounting for 15.5% of the global burden of disease[1,2]. A primary concern is that YP are at a point of transition that can often be challenging and confusing for them with vulnerabilities unique to their age[3]. Dissonance between culture and constructs of health service delivery can undermine and deter their service utilisation especially in highly sensitive and stigmatised areas such as sexual and reproductive health and mental health[4–6].

Research has been conducted on the factors that facilitate or hinder the uptake and utilisation of health services by YP, including the lack of trust in service providers due to power imbalances and negative attitudes held by health workers towards YP[7,8]. These factors create a gap between policy and practice at facility level[9–11].

South Africa's public healthcare system is structured in 5 layers to ensure cost effective access; (Primary Healthcare (Clinics), District hospitals; Regional hospitals, Tertiary (Academic) hospitals and Central (Academic) hospitals. Post 1994 transformation of the Department of Health involved creation of the national, provincial and local authority health departments[12]. The national department has overall responsibility for health care, while the provincial office provides and manages comprehensive health services through district delegation using the public healthcare model with a curative care-centred focus. The private sector facilities that guarantee health care through private practitioners and private hospitals tend to be in urban areas and serves 16% of the population[13]. Young people largely feel that they are not catered for within the health system and the Youth Friendly Services (YFS) programme is not implemented well enough especially in rural areas [14]. The young participants in this study are in a rural setting, accessing public sector health care. In this paper we argue that the behaviour, or the act of attending a health service is determined by beliefs and attitudes which are an antecedent of behaviour, while socioecological aspects shape this intention. We examine factors in the social environment that influence the constructs shaping behaviour from YP's perspectives in rural KwaZulu-Natal, South Africa.

### Theoretical concepts of behavioural change

Various social cognition principles have been applied to identify factors and mechanisms that influence behaviour, including Bandura's Social Learning Theory[15], the Health Belief Model

[16,17] and Theory of Reasoned Action[18,19]. Some of these models have been criticized as having limited evidence for long-term behaviour change, little or no predictive power, and discounting external influences e.g. cultural and socioeconomic factors that shape behaviour [20].

The Theory of Reasoned Action proposed that attitudes and subjective norms determine an individual's behavioural intention, and later evolved to the Theory of Planned Behaviour (TPB) with the addition of 'perceived control' as a third factor that influences behaviour. TPB has evidence for predictive validity and high accuracy in the causal links between intention factors and behaviour and has been successfully used in behavioural change interventions including smoking cessation, sexual transmitted infection testing, and uptake of exercise[21–23].

A critique of the social cognition models is the lack of cultural sensitivity with too much focus on the individual[24], suggesting the potential benefit of also considering the Socio-Ecological Model (SEM) which acknowledges many influences on health service utilisation behaviour[25]. The SEM framework portrays the multifaceted nature of personal and environmental factors that determine behaviours, and include five hierarchical levels (Individual, interpersonal, community, organizational, and policy/enabling environment (Fig 1).

Applying the TPB with a socio-ecological approach, may provide a pragmatic approach to afford insights on malleable constructs that predict behaviour.

In the context of the present study, service utilisation is motivated by a health-seeking behaviour preceded by intention and determined by a young person's attitude towards the use of a health service, (e.g. if I attend the clinic, I will receive appropriate health education that will improve my health). An individual can change, if s/he receives new information that can influence their behaviour[27]. For example, if a YP receives information about new opening times for a clinic, the YP will assess that information against their existing beliefs and make

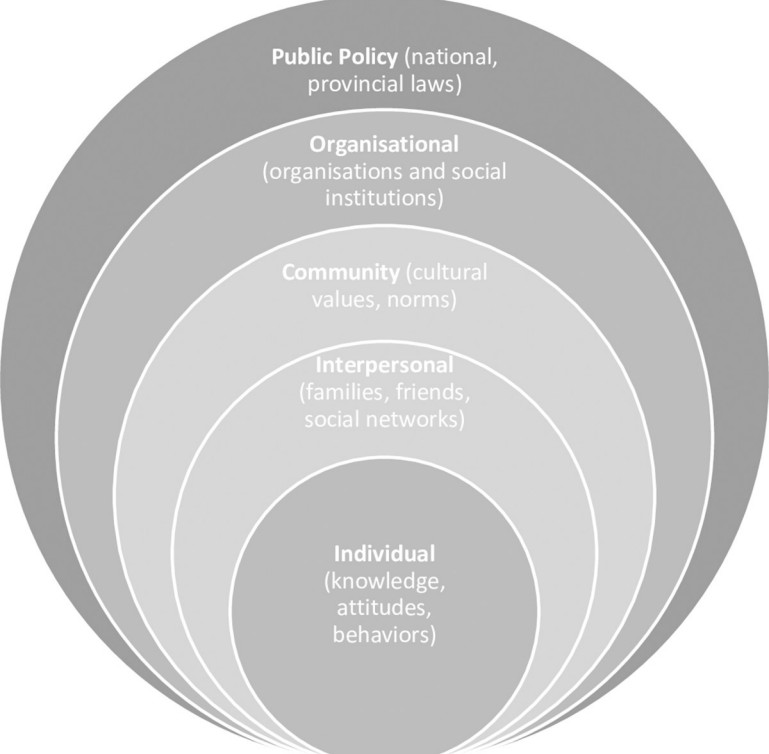

**Fig 1. The socio-ecological model[26].**

changes using this new information (i.e. to visit clinic at the new opening times). Intention is also influenced by the subjective norms shaped by important people around them such as friends, peers and family (e.g. If I go to the clinic my friend will think I have HIV) and by perceived behaviour controls (e.g. if there is a nurse that understands that I need to speak to them in confidence, I will go to the clinic).

In this paper we argue that the act of attending a health service is a behaviour and therefore an understanding of theoretical concepts of behavioural change is paramount in understanding health service utilisation. People's beliefs and attitudes determine intention to act which is an antecedent of behaviour [28] while socioecological aspects contribute to this intention. The overall aim of this paper which is from the qualitative data reported from a formative mixed-methods study, is to identify the beliefs that could be targeted in interventions. These data have helped in understanding factors in the social environment that influence the constructs shaping behaviour.

## Methods

### Study setting

The Health Services for young people study examined the utilisation of health care services by YP in a rural setting in Hlabisa and Mtubatuba sub-districts of uMkhanyakude district, Kwa-Zulu-Natal, South Africa. uMkhanyakude is one of the most deprived districts in South Africa with high HIV prevalence and high unemployment[29]. The district has five district hospitals, 57 clinics, including five gateway clinics, 17 mobile clinics servicing 238 mobile stopping points, and seven high transmission area (HTA) sites (two fixed and five mobile). The district has challenges of cross-border patients due to being on the border of Mozambique and Swaziland[30]. The setting is such that some young people walked up to 45 minutes one way to school, because the majority of the population live in scattered homesteads and a long distance from amenities. Services such as water and electricity are limited, and young people have few recreational facilities. Although the study area is largely rural, commercialisation has seen an increase in income-diversification as people try to move closer to the township, the local town, and the main transport routes to access work. This has impacted on proximity to healthcare services as people move to towns which takes them away from their allocated clinic. The political, economic and social legacies of apartheid affect the quality of primary care [30] with government-run public health clinics often poorly equipped, understaffed and do not offer adolescent friendly services.

### Participants and procedures

Two clinics were selected from the 17-primary health care clinics in Hlabisa and Mtubatuba sub-districts of UMkhanyakude, based on the clinic setting, ease of access to the facility, clinic headcount between April 2015 and February 2016. Community-based data were gathered from the study clinic catchment areas using various approaches as indicated in Table 1 which shows the participants that took part in each of the different methods.

This paper reports findings from: 1) Key Informant Interviews (KIIs) with professionals at the two primary health care clinics, one mobile clinic, school health team, two secondary schools, and one primary school; 2) exit interviews with YP (& their parent/guardian) attending health facilities; 3) In-Depth Interviews (IDIs) with YP in the community; 4) IDIs with community leaders, and 5) participatory mapping including a transect spiral walk followed by a focus group discussion (FGD).

A purposive sampling of convenience approach was taken. Purposive due to the research question as we sought the subjective experiences of young people in relation to health service access and utilisation. We therefore recruited young people who could share these experiences

**Table 1. Activities and participants in the study.**

| Activity | Male participants | Female participants |
|---|---|---|
| Clinic Exit Interviews with YP [interview conducted as a patient leaves the clinic to discuss their reasons for visiting the service obtained and their experience] | 3 | 5 |
| Accompanying person for exit Interview | 0 | 6 |
| In depth Interviews with community stakeholders | 5 | 10 |
| Key Informant Interviews with health facility staff | 3 | 3 |
| Key Informant Interviews with school professionals | 2 | 5 |
| In Depth Interviews with YP | 3 | 5 |
| Group discussion with school health team | 0 | 3 |
| Spiral transect walks followed by group discussions with YP | 6 (18–24 years)<br>5 (10–14 years)<br>6 (18–24 years)<br>5 (15–17 years) | 6 (18–24 years)<br>8 (10–14 years)<br>6 (15–17 years)<br>6 (10–14 years) |

and views as well as their guardians who are sometimes involved in their health care decisions. We recruited health care providers who attend to young people in the clinics and were therefore the most appropriate people to provide access to the information about youth health care utilisation. The convenience approach was due to the non-probability method where members of our study population were approached wherever we could find them, initially targeting places such as parks where young people gather. A team of 8 research assistants (4 male and 4 females) with a degree in a social science subject conducted data collection and were fully trained on the study protocol. For exit interviews, the research assistants had permission to approach young people as they left the clinic. The research assistants approached the clinics and schools and at each facility engaged with the head teacher or operations manager who then identified members of staff (Life Orientation teachers at school and youth ambassadors at clinics) that could take part in the study. The research assistant then approached the identified members of staff, introduced the research with full information given on the purpose of the study and proceeded to consent them if they showed an interest. There were 10 refusals, citing lack of time (n = 8), and concerns that the study would encourage undesirable sexual behaviours among AYP (n = 2).

Written informed consent was obtained from adults (aged 18 years and above). YP aged <18 years provided assent and parental/guardian consent was obtained. Ethical approval for the research was given by the University of KwaZulu-Natal Biomedical Research Ethics Committee (BREC) [BE472/15], and the Ethics Committee of the London School of Hygiene & Tropical Medicine [Ref 11092]. Permission from the AHRI Community Advisory Board was obtained as were approvals from the provincial, and district health officials.

## Data collection

Eight interviewers with a social science undergraduate degree collected data face to face from 101 participants over a 6-month period in 2017 using a topic guide that was piloted within the study setting. The interviewers had a two-week intensive qualitative skills and child protection training as they would be working with young people. They underwent further study specific training to equip them with appropriate skills to conduct the work. The main objectives of the interviews were to determine the availability of health care services for YP, to ascertain when and how they accessed these services, to understand the facilitators and barriers to service utilisation and how, from their perspective, services can be improved and this was communicated

to the participants during the recruitment and consent process. All interviews were conducted privately with only the participant and researcher present, digitally audio-recorded, with the individual's permission except for health care professionals where hand-written notes were requested in lieu of audio-recording. These notes were developed in extended interview scripts and the same analytical process was applied with the awareness that these scripts were not verbatim. Field notes were used to get an understanding of the setting and context of each data collection activity. Interviews ranged from 20 minutes (mostly exit interviews) to 60 minutes, while group activities (community mapping and group discussion) took up to 3 hours.

## Data analysis

Preliminary analysis was conducted using open coding following grounded theory approach. As a first step, each interviewer (8 interviewers) coded one transcript each with involvement and assistance by authors NN (PhD), BN (PhD) and LM (MA). Codes were discussed and combined into a codebook used to code the rest of the transcripts. The team met frequently to discuss the validity of new codes that emerged from individual coders. Data saturation was decided upon at the point when no new codes emerged from the data[31]. At the end of this initial process, the study team met to discuss the topics emerging from the indexing of the data, then identified and defined categories (broader themes).

The main categories were grouped systematically using the three TPB belief constructs, and an initial analytical framework was created. The analytical framework was used to identify indicators, constructs and characteristics associated with YP's behaviour. QSR NVivo 10 package was used to manage the data. The findings under each TPB construct were then collated and typed into an Excel file to be reviewed by the research team. For each belief construct, the researcher prepared an Excel file which outlined the categories identified within the belief, explanatory variables and evidence through quotations that show specific aspects of each variable. This review process was used to verify the associations made between the data and constructs of the TPB. After the initial review process, further interrogation of the data was done by grouping transcripts according to the data collection method (i.e. in-depth interviews, exit interviews, group discussions) and according to participant description (i.e. young person, health care professional, teacher etc.). This was done with the aim of identifying patterns across the dataset as suggested by Miles and Huberman[32]. Differences were settled through discussions until a consensus was reached.

Throughout, the interviewers who understood the context of the study setting were involved in the analysis. This minimised the insider-outsider effect of researcher positionality and was essential in interpreting the local dialect of the IsiZulu language. Other team members assisted in encouraging the interviewers to probe in depth so that important information or data were not lost. The interdisciplinary research team provided an outsider perspective as the non-local researchers were able to question some conclusions of the study bringing a different perspective. The different methods used also allowed us to triangulate the data and confirm information.

## Results

We present the findings aligned to the constructs of the Theory of Planned behaviour and the relevant socio-ecological element. The three constructs of TPB; 1) attitude towards behaviour (ATB) (both positive and negative attitudes); perceived norms (PN); and perceived control (PC) are used to frame the categories of the findings as shown in Fig 2. The results focus on behaviours, other structural factors that influence the use of services and showed a possible relation with TPB elements and the socio-ecological levels of influence.

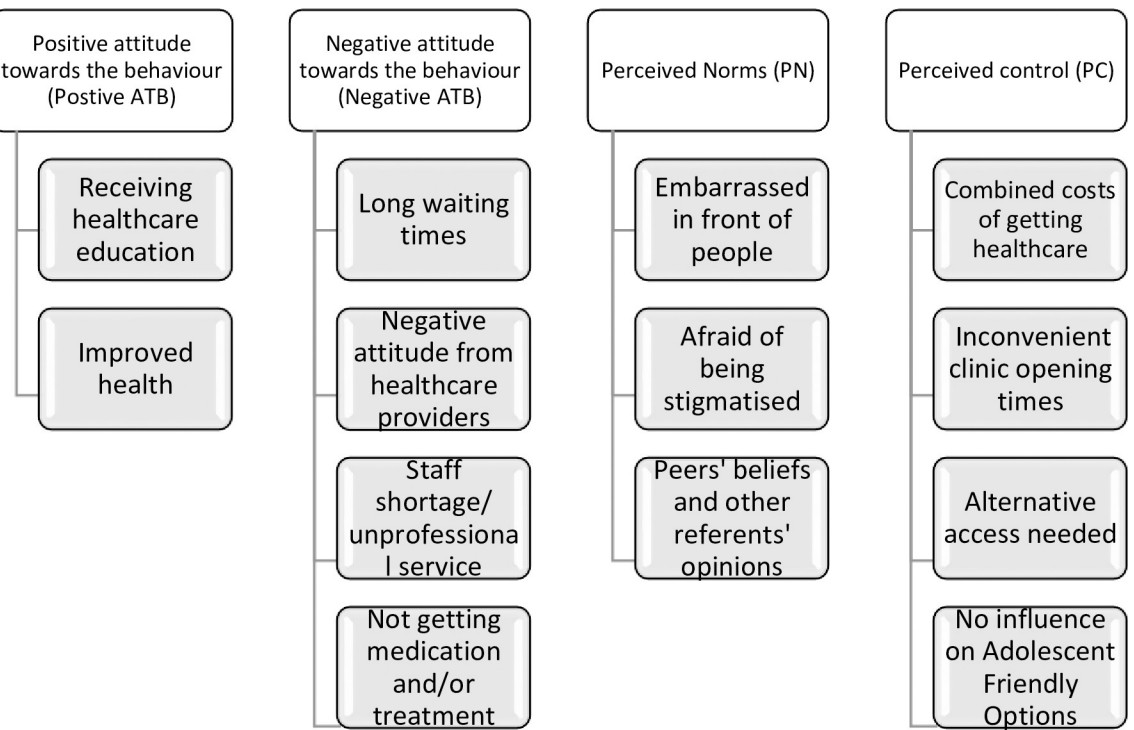

**Fig 2. List of accessible beliefs expressed by study population.**

## Positive attitude towards behaviour

Some YP identified positive attitudes towards health care services and facilities. Attitudes were linked to both the providers and the facilities available. YP were aware of the potential health benefits in attending the clinics such as improved health outcomes and acknowledged that clinics were a source for health education. A few YP also expressed that the healthcare workers were quite helpful towards them:

> "*It means that if you go to the clinic or the hospital they tell you, they tell us what food to eat*".
> [FGD with 18–24 years old men]

> "*They helped me, and they made me feel at ease because I knew that I would receive complete assistance.*" [IDI with 22-year-old man]

## Negative attitude towards behaviour

There were comments related to negative attributes related to care providers which were viewed as unprofessional, with nurses shouting at YP. This made young people less inclined to go to the clinic facilities. Another concern raised was that there were long waiting times due to the long queues. In comparison to visiting traditional healers, YP expressed that the health care facilities often had longer waiting times:

> "*If you get there [health facility] in the morning you wait until the sun set without getting any medication . . .In the traditional healer you get assistance at the same time you arrive there.*"
> [FGD with 15–17 years old boys]

Health care providers acknowledged the long waiting times and explained that this was due to staff shortages. Another concern shared by respondents was stock outs of medication. Some YP thought it was pointless to attend the clinic as there was a high chance of not getting medication they required:

"*Eh what they did, it was that they told me that the injections were out of stock and they gave me tablets that will fight the disease inside. . . What they can do is to make sure that medication is always available, maybe for the infections like Sexually Transmitted Infections.*" [Exit interview with 22-year-old man]

## Perceived norms (through social influence)

The fear of being stigmatised was expressed as a barrier to accessing services. Participants explained that in their community going to clinic for any healthcare service was often associated with Tuberculosis, HIV or other sexually transmitted infections (STI). This stigmatisation was even more pronounced for sexual and reproductive health services and led to gossip and stigma which made YP fear people's judgements and hence deterred services use:

"*. . .they (community) gossip about you and you sometimes end up quitting and go back home without acquiring what you have come for you are HIV positive and most people are aware about that, you would find that they will gossip and say bad things about you in the community. Maybe they tell other people to stay away from you. . .*" [FGD with 10–14 years old boys].

Some respondents also mentioned that they did not feel that, as YP, they were taken seriously. The clinics did not provide a safe space that gave their problems serious consideration. Instead they felt their questions were not appropriately answered:

"*I could be raped and be infected by certain diseases and also fall pregnant, but they wouldn't take me serious as a victim. . .They would just perceive it as we are lunatics . . .*" [FGD with 15–17 years old girls].

The impact of peers' beliefs was something that played a role in influencing whether to seek health care and how to do so. Peers were not the only referents that YP based their decision on, they also spoke of how the opinions and views of other important people such as family, friends and even the community influenced health seeking behaviour. An unexpected finding was that some YP said they preferred to share problems and seek advice from their mothers than friends:

"*If I have skipped my periods, I tell mom. . . If I tell my friends, they will gossip about my matter.*" [FGD with 18–24 years old women].

Another respondent explained how she discussed attending the clinic with her boyfriend. The girl explained what happened:

"*He refused when I told him that I want to start the antenatal visit at home and he said I must wait for him until he comes back home.*" [Exit interview with 20-year-old woman].

His opinion reflects a paternalistic view held by some in the area that the man is the decision maker.

## Perceived behaviour control

When asked what makes it difficult for them to attend the clinics, most responses related challenges posed by structural factors. The mapping exercise revealed few health facilities that YP felt they had access to within their local community. Participants expressed how the proximity of a clinic was a barrier:

"*If clinic can be closer because the clinic is too far others are even failing to have passion of going to clinic . . .I can't be able to walk it's too far.*" [IDI, 55-year-old woman]

Service availability was described in terms of opening times with reports on a need for clinics to have more convenient opening times. YP explained that sometimes they do not go to the clinic when they are unwell as they are meant to be in school. They reported that sometimes healthcare professionals do not implement the government initiative to fast-track students and so if they do decide to access services, then they are forced to skip school:

"*The reason for my visit today was due to illness, which I think started maybe over a week ago. . . and I decided to skip school and leave home to visit the health care facility and seek a medical intervention for this illness that I am suffering from for all this time.*" [IDI 22-year-old man].

Many YP expressed the need for easy access to healthcare services through initiatives that would motivate, empower them with the education and skills to take care of themselves:

"*What can be done to help young people, is if in different areas. There can be older people who can collect young people and give health education maybe that can keep them together.*" [FGD with 18–24 years old men].

"*If we can have groups in our community like cooperatives to work together and take care of our area. . . health care givers like buddies who will do home visits . . .*" [FGD with 18–24 years old women].

## Discussion

Based on the TPB and the findings from this formative work, exploring and identifying modifiable belief constructs can inform possible interventions needed to promote service utilisation by YP, while the SEM identifies the levels where these interventions can be targeted as shown in Fig 3 below.

### Individual and behavioural beliefs

At an individual level of the SEM, the construct that seemed to emerge as playing a bigger influence was behavioural beliefs of young people. Although, young people mentioned that they received health education and good health outcomes from accessing care at the clinic, the negative aspects including/such as were expressed more often than the positive factors. This has important implications on young people's acceptance of interventions as beliefs play an important role in the intention to carry out a behaviour. If someone has strong beliefs about their ability to perform a certain behaviour, this increases their self-efficacy and the likelihood that the behaviour will be performed even when there are other barriers in the way[19]. This suggests that interventions that focus on the individual level of SEM by modifying the current negative beliefs that young people have about services and facilities may be effective. It is

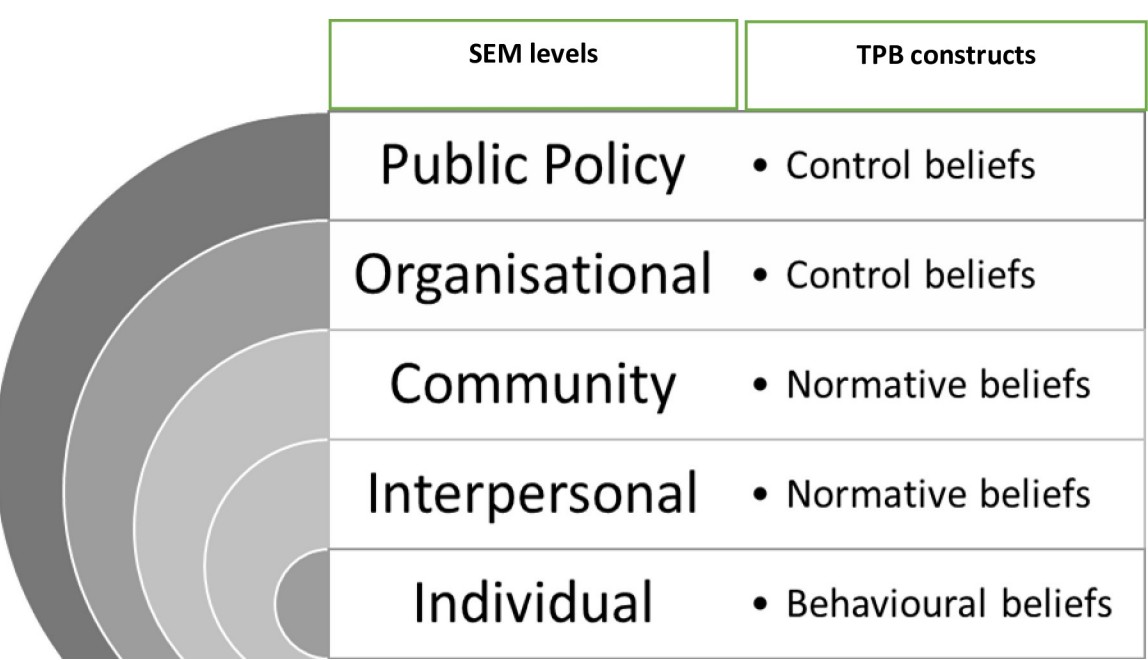

**Fig 3. Identified socio-ecological levels of influence with associated theory of planned behaviour belief constructs.**

therefore essential to consider factors and processes relevant to the individual, as well as the disjuncture between community level cultural values, norms and national policy in the effective implementation of strategies and programmes.

### Interpersonal, community and normative beliefs

This study shows that the TPB constructs are also determined by underlying beliefs that may be influenced by the environmental context at the interpersonal and community level of SEM. Negative beliefs seem to be largely influenced by aspects or issues that occur at the community, organisational and public policy level of the SEM. At the interpersonal and community levels, participants were influenced by peers, family as well as friends which contributed to their normative beliefs evoking the heightened need for privacy and confidentiality. A lack of privacy and confidentiality contributed to young people avoiding the clinics in fear of being stigmatised within their communities. This is consistent with previous research which has shown stigma to be a barrier to engaging in services[33]. One unanticipated finding was that young people sought health advice from their mothers. This corroborates a statement from the WHO quality standards for adolescent-friendly health services that sometimes young people may prefer to seek help from their mothers before approaching a nurse or clinic[34]. This has important implications on the involvement of the whole community in developing services and to engage parents/guardians to encourage AYP to access services when they need to and speaks directly to the community level of the SEM in designing interventions that are culturally tailored to that community.

### Organisational system/level and control beliefs

At the organisational level, participants continuously emphasised the importance of a favourable environment in the health service facility and easy access. These findings support previous research which identifies accessibility, staff characteristics and lack of confidentiality as

indicators specific to youth needs, that must be addressed in developing youth friendly services [35–37]. The environment of a health facility contributed to control beliefs that one held. Negative attributes of these beliefs focused on the difficulty in accessing care due to structural and socioeconomic factors such as lack of transport to get to the clinic, lack of trust in the competency and professionalism of care providers due to staff shortages as well as scarcity of medicines. These findings are in line with those of Hardin and colleagues who found that young people had lower trust in healthcare providers compared to adults[8]. This also highlights the need for care providers to be upskilled in adolescent care and communication which may break down this barrier experienced by youth [35,38]. Young people feared that their problems would not be taken seriously, and they would not receive answers to their questions. This demonstrates the need for an organisational level intervention in providing communication and competency training for care providers who may not recognise that their attitude may negatively influence the intention to utilise health services for young people. In addition, the findings support principles of youth friendly services and evidence from a WHO systematic review of the effectiveness of interventions to improve the use of health services by adolescents in developing countries which recommends that health care workers should be non-judgmental and ensure that their beliefs and views do not negatively impact the service they provide[39] Adolescent training for health care providers was identified as an important aspect in developing effective and friendly services[40].

## Public policy and control beliefs

At the higher public policy level of SEM, the Youth Friendly Health Service (YFHS) policies and guidelines have identified factors that should be taken into consideration in developing youth friendly services. Several of these guidelines are closely related to the list of accessible modal beliefs identified in this study which focus on acceptability, accessibility, and appropriateness of services. This, however, does not indicate that all the youth friendly health services have the same level of effectiveness in all regions. The WHO standards report advises that the implementation should be based on information derived from certain steps that inform which guidelines are needed and/or appropriate for the setting [40]. In line with the WHO steps in developing and implementing YFHS, this study provides a good understanding of the current context in which services are provided and their utilisation by young people as well as the programmatic opportunities and challenges.

## Implications/Lessons for interventions targeting YP

At an exploratory level, this study has identified potentially modifiable determinants of behaviour that play a role in influencing intentions such as the introduction of information addressing perceived norms that can lead to the formation of new beliefs in YP. Although discussed by a few participants, one such example suggested by participants would be introduction of new information through a peer or buddy system. Buddy programmes have been shown to be effective in promoting service utilisation and retention in care and in promotion of positive healthcare[41,42]. In South Africa, these peer-led programmes are largely initiated and run by NGOs working on strengthening systems and HIV prevention and work collaboratively with the Department of health (DoH) as implementing partners[43]. Peer to peer interventions have been recommended as a way to upscale support platforms as part of South Africa DoH's Adolescent and Youth Health Policy 2016–2020[44]. Another strategy that was recommended by the YP is having adult mentors to empower them through health education to increase their health literacy.

Further work can be done to quantitatively measure the relative contributions of each of these factors to intention and each belief's variability to give more insight into the levels and weights of predictor variables that an intervention would need to target. The greater the weight of a variable, the greater the chances of influencing behavioural change albeit with the awareness that utilisation of services as a behaviour is complex and needs to be addressed at various SEM levels.

### Strengths and limitations

This project involved a wide variety of stakeholders with multi-methods approaches in order to engage YP. Another strength was the use of local interviewers who could speak the local dialect. Their involvement in preliminary analysis ensured that the findings were grounded in the data and reduced misinterpretation bias. The use of theory to inform analysis helped to make a link from abstract to organised explanations giving a broader understanding and significance of the data. The study had some limitations. The sample was purposely selected and may therefore not be representative of the general population of YP in the local area. The size of the sample is an indication of the explorative nature of the work; however, this does not reduce the importance of the factors identified that impact health care utilisation. During exit interviews at the clinic, we could not recruit YP under the age of 18 years unless they had an accompanying adult with them and many YP less than 18 years of age attended clinics on their own or with friends. Due to the setting of the Institute, local power relations and respondents' understanding of research purposes, it is possible that negative experiences were more likely to be reported than positive experiences. However, the interviewers were local YP who used the local dialect and were immersed in the culture of the participants.

### Conclusion

Theories of health behaviour can be useful tools in identifying both sources of influence on behaviour and interventions to target those influences. TPB is a useful framework for identifying variables that need to be modified in changing service utilisation behaviour, while SEM shows that there are multiple levels that influence behavioural interventions. Identifying the belief constructs and their associated SEM level, helps in understanding that targeting multiple levels of the framework could be more effective. This comprehensive framework incorporates individual cognitive aspects and environmental components which need to be addressed to change behaviour and promote service utilisation. Despite its exploratory nature, this study offers insight into potential interventions at different levels of the socio-ecological model. The approach of using both the theory of planned behaviour and the socio-ecological model highlights the need for interventions to be culturally tailored to the context.

### Acknowledgments

We acknowledge Africa Health Research Institute's Community Advisory Board that gave critical feedback on this project as well as appropriate methods for use within the community. We would like to acknowledge the KwaZulu-Natal uMkhanyakude district Department of Health for the support and governance permissions given to conduct the study. We would also like to thank our data collection team and all the study participants for their time.

### Author Contributions

**Conceptualization:** Jane Ferguson, Janet Seeley, Aoife M. Doyle.

**Data curation:** Nothando Ngwenya, Busisiwe Nkosi, Lerato S. Mchunu, Aoife M. Doyle.

**Formal analysis:** Nothando Ngwenya, Busisiwe Nkosi, Lerato S. Mchunu, Janet Seeley, Aoife M. Doyle.

**Investigation:** Nothando Ngwenya, Lerato S. Mchunu, Aoife M. Doyle.

**Methodology:** Janet Seeley, Aoife M. Doyle.

**Project administration:** Nothando Ngwenya.

**Writing – original draft:** Nothando Ngwenya, Janet Seeley, Aoife M. Doyle.

**Writing – review & editing:** Nothando Ngwenya, Busisiwe Nkosi, Lerato S. Mchunu, Jane Ferguson, Janet Seeley, Aoife M. Doyle.

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
