## [Decision Letter · Decision Letter 0]

14 Feb 2020

PONE-D-19-19930

Behavioural and socio-ecological factors that influence access and utilisation of health services by young people living in rural KwaZulu-Natal, South Africa: Implications for intervention

PLOS ONE

Dear Dr Ngwenya,

Thank you for submitting your manuscript to PLOS ONE. After careful consideration, we feel that it has merit but does not fully meet PLOS ONE’s publication criteria as it currently stands. Therefore, we invite you to submit a revised version of the manuscript that addresses the points raised during the review process.

I received very positive feedback and I have carefully reviewed the manuscript as well. Overall, the reviewing team thought that the study was well-articulated and stayed quite focused on its theoretical framework which greatly contributed to its clarity and impact. I would ask you to please consider the reviewer’s minor comments in a revision.

We would appreciate receiving your revised manuscript by Mar 30 2020 11:59PM. To enhance the reproducibility of your results, we recommend that if applicable you deposit your laboratory protocols in protocols.io, where a protocol can be assigned its own identifier (DOI) such that it can be cited independently in the future. For instructions see: http://journals.plos.org/plosone/s/submission-guidelines#loc-laboratory-protocols

We look forward to receiving your revised manuscript.

Kind regards,

Rachel A. Annunziato, Ph.D.

Academic Editor

PLOS ONE

Journal Requirements:

2. Please change "female” or "male", when used as a noun, to "woman”, "man", "boy", "girl" or other such terms, as appropriate.

3. In the ethics statement in the Methods and online submission information, please ensure that you have specified, both for participant consent and parental consent, (1) whether consent was informed and (2) what type you obtained (for instance, written or verbal, and if verbal, how it was documented and witnessed).

5. We note you have included a table to which you do not refer in the text of your manuscript. Please ensure that you refer to Table 1 in your text; if accepted, production will need this reference to link the reader to the Table.

Reviewers' comments:

Reviewer's Responses to Questions

**Comments to the Author**

1. Is the manuscript technically sound, and do the data support the conclusions?

Reviewer #1: Partly

2. Has the statistical analysis been performed appropriately and rigorously? 

Reviewer #1: Yes

3. Have the authors made all data underlying the findings in their manuscript fully available?

Reviewer #1: Yes

4. Is the manuscript presented in an intelligible fashion and written in standard English?

Reviewer #1: Yes

5. Review Comments to the Author

Reviewer #1: The manuscript is technically sound scientific research with robust data points that support the conclusions. The use of both quantitative and qualitative data presents a comprehensive exploration of health service utilization by young people in rural KwaZulu-Natal, South Africa.

Statistical analysis has been performed appropriately. However, I was unable to locate the summarizing statistics in Figures 1-3. These figures did not seem to appear in the manuscript and would have greatly contributed to my understanding of the statistical analysis and data visualization tools used.

Language in manuscript is clear, correct, and unambiguous. Results are presented in an intelligible fashion that is also accessible to non-experts.

6. PLOS authors have the option to publish the peer review history of their article (what does this mean?). If published, this will include your full peer review and any attached files.

Reviewer #1: No

---

## [Author Response · Author response to Decision Letter 0]

29 Feb 2020

Response: We have followed PLOS ONE’s publishing style requirements and checked all formatting against the templates provided. 

2. Please change "female” or "male", when used as a noun, to "woman”, "man", "boy", "girl" or other such terms, as appropriate.

Response: Thank you for the suggestion. We have made the necessary changes in the instances where ‘female’ or ‘male’ was used as a noun and removed other instances for readability.

3. In the ethics statement in the Methods and online submission information, please ensure that you have specified, both for participant consent and parental consent, (1) whether consent was informed and (2) what type you obtained (for instance, written or verbal, and if verbal, how it was documented and witnessed).

Response: All the consent was informed, as well as assent for minors. Only written informed consent was obtained from all participants as well as from parents in the case of minors under the age of 18 years old. Verbal consent was not used in this research.

Response: There are ethical restrictions on sharing these data publicly due to data containing potentially identifying participant information. 

5. If there are ethical or legal restrictions on sharing a de-identified data set, please explain them in detail (e.g., data contain potentially sensitive information, data are owned by a third-party organization, etc.) and who has imposed them (e.g., an ethics committee). Please also provide contact information for a data access committee, ethics committee, or other institutional body to which data requests may be sent.

Response: We have indicated this in the Data availability statement, “Data cannot be shared publicly because of confidentiality and potential breach as the data contains potentially identifying participant information. Data are available from the AHRI Research Data Management committee (contact via RDMServiceDesk@ahri.org) for researchers who meet the criteria for access to confidential data.”

6. If there are no restrictions, please upload the minimal anonymized data set necessary to replicate your study findings as either Supporting Information files or to a stable, public repository and provide us with the relevant URLs, DOIs, or accession numbers. For a list of acceptable repositories, please see http://journals.plos.org/plosone/s/data-availability#loc-recommended-repositories.

Response: “Data cannot be shared publicly because of confidentiality and potential breach as the data contains potentially identifying participant information. Data are available from the AHRI Research Data Management committee (contact via RDMServiceDesk@ahri.org) for researchers who meet the criteria for access to confidential data.”

7. We note you have included a table to which you do not refer in the text of your manuscript. Please ensure that you refer to Table 1 in your text; if accepted, production will need this reference to link the reader to the Table.

Response: Thank you for this suggestion and we have updated the participants and procedures section pg 8, lines 148 – 150 to help make an easier reference to the Table. 

Review Comments to the Author

8. Reviewer #1: The manuscript is technically sound scientific research with robust data points that support the conclusions. The use of both quantitative and qualitative data presents a comprehensive exploration of health service utilization by young people in rural KwaZulu-Natal, South Africa.

Response: We thank the reviewer for their comment and appreciation of this manuscript being technically sound scientific research. 

9. Statistical analysis has been performed appropriately. However, I was unable to locate the summarizing statistics in Figures 1-3. These figures did not seem to appear in the manuscript and would have greatly contributed to my understanding of the statistical analysis and data visualization tools used.

Response: Figures 1-3 were a visualisation of the theoretical frameworks that were applied to assist with the data analysis as well as the intersection and interaction of the theories used. We found this useful in describing the pathway we took in conducting the analysis and shows the analytical framework used to identify indicators, constructs and characteristics associated with young people’s behaviour in our research community.

10. Language in manuscript is clear, correct, and unambiguous. Results are presented in an intelligible fashion that is also accessible to non-experts.

Response: We thank the reviewer for this positive comment as we tried to ensure that the content, especially that of the theoretical underpinning was accessible to non-experts.

---

## [Editor Report · Decision Letter 1]

17 Mar 2020

Behavioural and socio-ecological factors that influence access and utilisation of health services by young people living in rural KwaZulu-Natal, South Africa: Implications for intervention

PONE-D-19-19930R1

Dear Dr. Ngwenya,

We are pleased to inform you that your manuscript has been judged scientifically suitable for publication and will be formally accepted for publication once it complies with all outstanding technical requirements.

With kind regards,

Rachel A. Annunziato, Ph.D.

Academic Editor

PLOS ONE
---

## [Editor Report · Acceptance letter]

24 Mar 2020

PONE-D-19-19930R1 

Behavioural and socio-ecological factors that influence access and utilisation of health services by young people living in rural KwaZulu-Natal, South Africa: Implications for intervention 

Dear Dr. Ngwenya:

I am pleased to inform you that your manuscript has been deemed suitable for publication in PLOS ONE. Congratulations! Your manuscript is now with our production department. 

With kind regards,

on behalf of

Dr. Rachel A. Annunziato 

Academic Editor

PLOS ONE